# Development of UVB LED Lighting System Based on UV Dose Calculation Algorithm to Meet Individual Daily UV Dose

**Dae-Hwan Park [1], Seung-Taek Oh [2] and Jae-Hyun Lim [1,\*]**

[1] Department of Computer Science & Engineering, Kongju National University, Cheonan 31080, Korea; glow153@smail.kongju.ac.kr
[2] Smart Natural Space Research Center, Kongju National University, Cheonan 31080, Korea; ost73@kongju.ac.kr
\* Correspondence: defacto@kongju.ac.kr; Tel.: +82-10-8864-6195

**Abstract:** Ultraviolet B (UVB) in sunlight is known to promote health when humans are exposed to optimum sunlight. Proper exposure to ultraviolet B is essential to produce vitamin D in the body, which is a particularly important factor for health. However, there has been an increase recently in the number of people who are lacking sunlight exposure due to staying indoors. Avoiding ultraviolet (UV) rays leads to health deterioration. To solve this problem, a portable ultraviolet measuring device that provides users with the UV intensity information of outdoor sunlight has been developed while UVB Light Emitting Diode (LED) lighting technologies capable of providing UVB radiation have been studied. However, existing technologies are mere methods for providing information on ultraviolet rays and artificially exposing to ultraviolet rays, and there is a risk for the UV radiation amount to not meet the daily required UV dose. Therefore, in this paper, a UVB LED general lighting system based on a UV exposure calculation algorithm that supports users' daily required UV dose is proposed. The proposed system is composed of an IoT (Internet of Things) UV measuring device that measures the UV dose indoors, as well as outdoors, UVB LED general lighting which can safely provide UVB doses at indoors, and a smartphone which provides information on the cumulative UV dose and the estimated amount of vitamin D while it controls UVB LED general lighting. In addition, it is possible to support users' vitamin D synthesis by providing as much UV light as its shortage through UVB LED general lighting, based on an individual's UV exposure amount through outdoor sunlight, using a smartphone implementing a UV dose calculation algorithm. In order to confirm the performance of the proposed system, a simulation was conducted assuming that users with skin types 2 and 3 performed outdoor activities within 30 min and entered the room. The result confirmed that the proposed lighting can provide users of all ages with the UV dose required per day.

**Keywords:** UVB LED general lighting system; vitamin D UV dose calculation algorithm; daily recommended vitamin D UV dose; UVB LED; UV measurement device

---

## 1. Introduction

Since natural light contributes to the prevention of disease and the suppression of mental illness, proper exposure to daylight is an essential element for enhancing physical and mental health [1]. Natural light includes infrared and ultraviolet (UV) rays that are not provided by general lighting, and these elements have a great impact on human health [2,3]. Particularly, UV light may cause harmful or beneficial effects depending on a person's degree of exposure; therefore, precaution is needed for appropriate exposure [4]. UV is divided into ultraviolet A (UVA) (315–400 nm) and ultraviolet B (UVB) (280–315 nm), which reach the earth surface, and ultraviolet C (UVC) (200–280 nm), which is mostly

absorbed in the ozone layer and cannot reach the surface. UVA penetrates deeper into the dermal layer of the skin and produces reactive oxygen species that cause skin aging and wrinkle formation [5,6]. Meanwhile, UVB causes skin pigmentation, weakens skin barriers, and promotes erythema and skin cancer [7,8]. In general, UV is known to be harmful to human body, but UVB helps the body synthesize vitamin D which is an essential element for human health, benefiting health by relieving hypertension and cardiovascular disease [9,10]. Particularly, vitamin D in the body is supplied in a very small amount through food, and most of it is known to be synthesized in the body after UVB in the sun is irradiated to the skin. Therefore, appropriate exposure to UV has recently been recommended [11,12]. However, it is becoming difficult for modern people to be exposed to natural light due to increased indoor dwelling and UV exposure insufficiency. Kecorius has reported that people spend an average of 84% of their time indoors during the day [13], and Duarte has reported that exposure to UVB is limited at indoors since UVB cannot penetrate general glass [14]. As exposure to natural light and UV rays becomes difficult, various studies and efforts are being made to provide UVB indoors. In some research papers, recommended time and methods for optimum exposure to UV according to skin type, age, and current UV index (UVI, Ultraviolet index) are suggested [15], and a portable measuring device that provides information about UV during outdoor activities and consequent estimated amount of vitamin D synthesis was developed [16]. However, such methods only either suggest to the user a duration for outdoor activity or provide passive information about estimated amounts of UV and vitamin D synthesis. On the other hand, Chandra verified the vitamin D synthesis effect on vitamin D deficient patients by using a commercial tanning device with an artificial UV light source, and a UV special lighting device was developed that provides a UVB dose to parts of the body for skin treatment [17,18]. Research has also been carried out on the development of UVB Light Emitting Diode (LED) indoors lighting, which provides harmless levels of UVB light [19]. However, these UV lighting technologies control the irradiation time of lighting among the users, but they fail to implement a lighting system that provides UV dose at indoors by linking to the UV exposure amount of an individual outdoors.

Therefore, this paper proposes a UVB LED general lighting that provides users with a satisfactory UV dose required daily by individuals based on the calculation algorithm of UV exposure amount. The proposed system consists of an IoT (Internet of Things) UV measuring device, a smartphone, and UVB LED general lighting. The IoT UV measuring device measures the strength of UVI when the users conduct outdoor activities and sends the result to the smartphone. In the smartphone, cumulative UV doses are calculated based on the received UVI values and an output of daily optimum UV dose is given considering the skin type of the user, age of the user, and exposed area. When the cumulative UV dose of the user cannot reach the daily optimal UV dose, it compensates for the insufficient UVB dose through UVB LED general lighting. At this time, the UV dose of UVB LED general lighting is understood to be the safest level against the evaluation level of UV range in the photobiological safety standard of the lighting. After that, it supports the daily optimal exposure to UV of the user by providing the lacking UVB dose during outdoor activities through UVB LED general lighting indoors. In addition, a simulation is performed considering the skin type of the user and duration of outdoor activities (0 to 30 min) to validate the performance and practicality of the proposed system.

## 2. UVB LED General Lighting System

UVB LED general lighting system measure the users' UV exposure amount during outdoor activities. In case the exposure amount to UV by the users cannot meet the daily optimal UVB dose, the UVB dose is provided through UVB LED general lighting installed indoors as compensation to the shortage of UV exposure amount. Figure 1 shows the conceptual drawing of the UVB LED general lighting system. The proposed system was composed of an IoT UV measuring device that measures UVI outdoors, UVB LED general lighting that provides UVB doses indoors, and a smart phone application that provides information to the users such as UVI and vitamin D synthesis amount and that controls UVB LED general lighting. First, the IoT UV measurement device acquired UVI data in real time, calculated erythemal UV (EUV), and sent the result to the smartphone application

through Bluetooth Low Energy (BLE) communication. In the smartphone application, information on the expected amount of vitamin D synthesis by real-time UVI and outdoor activities was calculated and provided to the user. In addition, the presence of the user was judged by checking UVB LED general lighting through regular scanning. When it was judged that the user had come in indoors, the UVB LED provided the short UVB dose throughout the room to meet the daily recommended UV dose.

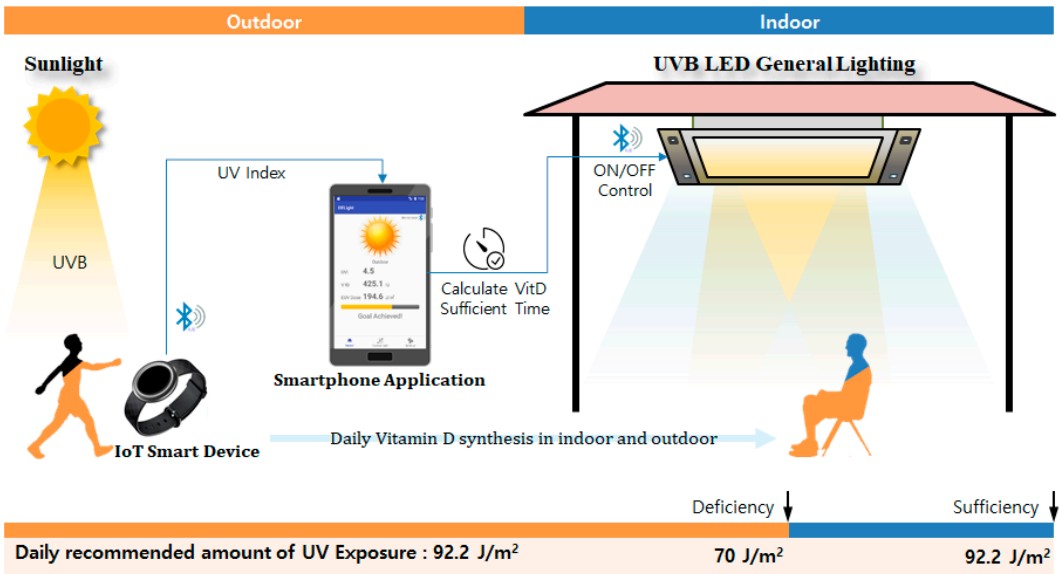

**Figure 1.** Conceptual drawing of the UVB general lighting system.

*2.1. System Construction*

The UVB LED general lighting system proposed in this paper is comprised of an IoT smart device, a smartphone, and UVB LED general lighting as shown in Figure 2. The IoT UV measuring device was constructed with a UVI sensor that measures real-time UVI outdoors, an Arduino-based microcontroller (MCU) that performs UVI calculation and communication functions, and a BLE module that sends the measured UVI information to the smartphone. The smartphone application was used to calculate the daily cumulative UV dose of the user, the estimated vitamin D synthesis amount, and the control time calculation of the UVB LED lighting to meet the insufficient amount of UV dose. In addition, a setting function was used to set the user's information so that the optimal UV dose required for a day according to the ages of the users can be calculated differentially. Furthermore, a visualized service was implemented to provide users with the communication function for data sending and receiving between the IoT UV smart device and UVB LED lighting, as well as UV and vitamin D related information. The UVB LED general lighting was constructed with the UVB LED, visible LED, and BLE module. In addition, a UVB LED lighting module was implemented to perform a communication function to receive the lighting control information from the smartphone and to operate a UVB LED lighting module based on the received information so that the UVB lighting service can be executed indoors.

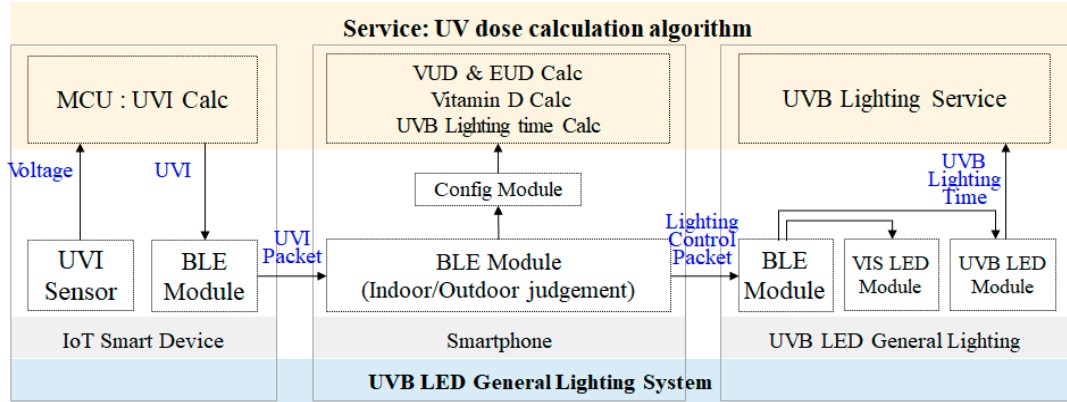

**Figure 2.** Construction of UVB general lighting system.

## 2.2. IoT UV Measuring Device

An IoT UV measuring device was constructed with the BLE module for wireless communication with the UVI sensor to measure the UV and microcontroller unit (MCU) and to carry out the measurement and data sending of UVI. The implemented UVI sensor was from Tocon E2 (Sglux, Germany) which discharges output voltage according to the strength of the UV index. In addition, ATmega328 (Atmel, San Jose, USA) was implemented for the MCU. The packet for communication was then constructed while sending functions through the BLE module (HM-10). Figure 3 shows the construction and process flow of the IoT UV measuring device.

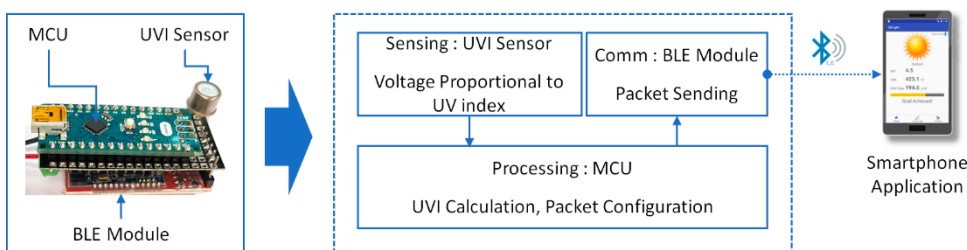

**Figure 3.** Block diagram and process flow of Internet of Things (IoT) ultraviolet measuring device.

UVI is expressed as Equation (1) by quantifying the relative intensity of UV as an arithmetic indicator by the World Health Organization (WHO) [20]. Additionally, the UVI sensor (Tocon E2) returns UVI, which is the integrated value for each wavelength in Equation (1). Therefore, EUV, which is essential for the calculation of the amount of vitamin D synthesis, was calculated by applying the UVI value collected through the sensor to Equation (1).

$$UVI = 40 \int_{280}^{400} E(\lambda) \cdot W_e(\lambda) \, d\lambda = 40 \, [\text{m}^2/\text{W}] \cdot EUV \, [\text{W}/\text{m}^2] \tag{1}$$

In Equation (1), $\lambda$ is the wavelength, $E(\lambda)$ is the spectral irradiance of the wavelength $\lambda$, and $W_e(\lambda)$ is the action spectrum for erythema for wavelength $\lambda$. The communication function was implemented to send real-time UVI measurement results to the smartphone.

## 2.3. UVB LED General Lighting

UVB LED general lighting was developed in the form of general lighting to allow the user to be exposed to UVB naturally during everyday living. Figure 4 shows the overall configuration of the UVB LED.

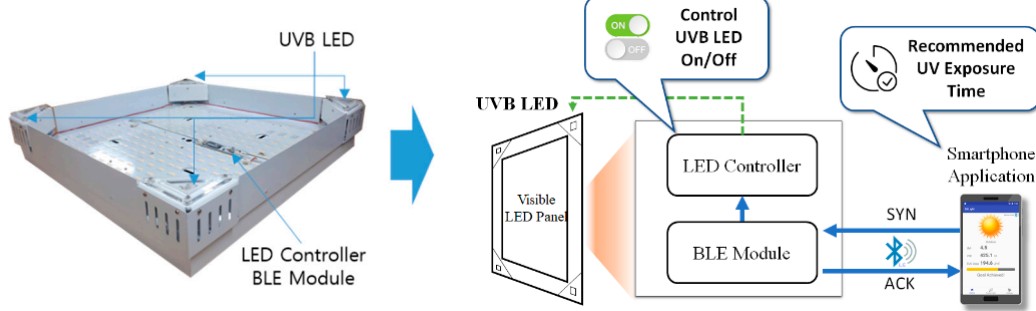

**Figure 4.** UVB LED general lighting and block diagram of UVB LED general lighting.

For UVB LED general lighting, the visible LED was arranged broadly at the center of the room to provide the optimum illumination. At this time, the visible LED was arranged in such way that it could realize the recommended illuminance of 500 lux according to ISO 8995-1 at a distance of 150 cm [21]. In addition, a set of LEDs was composed of visible LED chip of four types, two for Warm white and two for Cool white (Warm LED: 2600 and 3800 K, Cool LED: 5000 and 6400 K) and LEDs were arranged by 16 sets horizontally and 18 sets vertically on the visible LED panel. The lighting was designed to adjust brightness and color temperature as needed. In addition, four UVB LED (LG InnoTek, Korea) light sources were implemented on the corners to provide a UVB dose. The applied UVB LED light source had the rated current of 350 mA, rated voltage of 8.5 V, power consumption of 4250 mW, and radiant flux of 20 mW. In addition, the functions of receiving an SYN packet that contained the control time information of the lighting from the smartphone application, and sending of the ACK packet that informed normal receiving of the lighting control time data to the smartphone application, were implemented in the BLE module. Furthermore, the ON/OFF control function of the UVB LED lighting was implemented.

*2.4. Smartphone Application*

In the smartphone application, a communication function for data transmission between IoT UV device and UVB LED general lighting and a setting module for entering user's age was implemented. In addition, the smartphone application was made to provide the information on UV and the estimated vitamin D synthetic amount to the users. In order to promote the safe use of UVB LED lighting and to avoid overexposure to UV, an alarm function was provided when the minimum EUV dose (MED), which is the EUV dose causing skin erythema, was exceeded. MED was set to 250–350 J/m$^2$ for skin type II and 300–500 J/m$^2$ for skin type III [22]. Furthermore, the IoT UV device and UVB LED, the communication module for sending and receiving data between indoor and outdoor lighting and the setting module for inputting user's age information, respectively, were also implemented. For the communication protocol for the jobs, BLE was used and it was always connected with the IoT UV measuring device. Indoors and outdoors could be distinguished by sensing the presence of UVB LED general lighting through BLE scanning. The scanning by the BLE device was continuously performed and if UVB LED lighting was sensed, it was considered that the user entered the room. A connection between the smartphone and UVB LED lighting was also established. Figure 5 shows an example of the notification function of an Android OS based mobile app that implements the daily UV dose calculation function. Figure 5a shows the screen that provides UVI information which is received from the IoT UV measuring device and the Erythemal UV Dose (EUD) calculated based on that information. In addition, Figure 5b shows the values of control time of UVB LED general lighting according to the satisfaction of the daily optimal UVB dose when the user is situated indoors. Information on the amount of estimated amount of vitamin D synthesis to date and information on vitamin D producing UV radiation (Vitamin D UV Dose, *VUD*) being cumulated by UVB LED general lighting is also displayed. Figure 5c shows the results when the daily recommended vitamin D UV

dose (*RVUD*) is satisfactory through exposure to the sunlight and indoor UVB LED lighting. Figure 5d shows an example of the alarm function provided when the *EUD* in and out of the room exceeds MED.

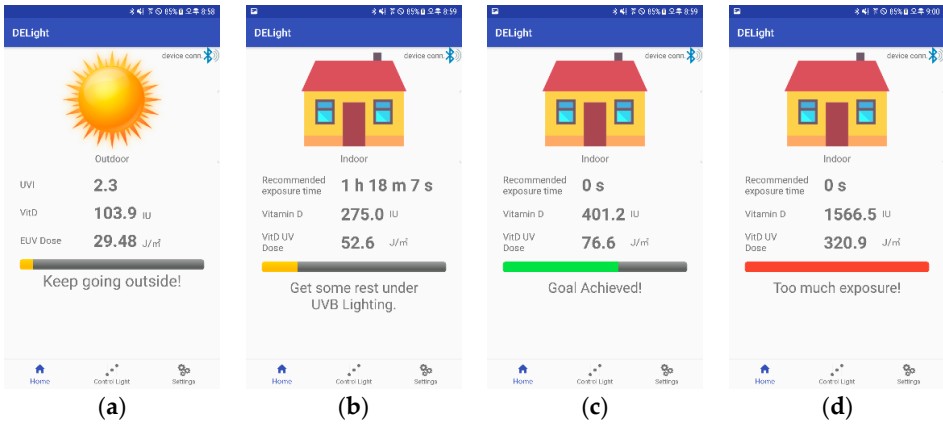

(**a**)    (**b**)    (**c**)    (**d**)

**Figure 5.** Mobile App on Notice: (**a**) outdoor; (**b**) indoor; (**c**) indoor, vitamin D sufficient; (**d**) when erythemal UV dose exceeds minimum erythemal UV dose.

## 3. UV Exposure Amount Calculation Algorithm

In this paper, a function was developed to provide the necessary UV dose through the indoor UVB LED lighting based on the UV exposure amount by sunlight in order to support the optimal daily UV dose required by the user. The UV exposure amount calculation algorithm calculates the optimal UVB dose necessary for a day according to the characteristics of an individual, such as the age and skin type of the user, performs smartphone-based BLE scanning, and then searches UVB LED lighting to judge whether a user is indoors or outdoors at present. Outdoors, the lighting system judges if the UV dose is optimal to the individual according to the cumulative EUV by exposure to the sunlight. Indoors, if it judges that the UV exposure amount of a subject is short while outdoors, the irradiation time by indoor UVB lighting is calculated to meet the daily UV dose by individual, and then the appropriate UVB dose is provided. Figure 6 shows the flow chart of the UV exposure amount calculation algorithm.

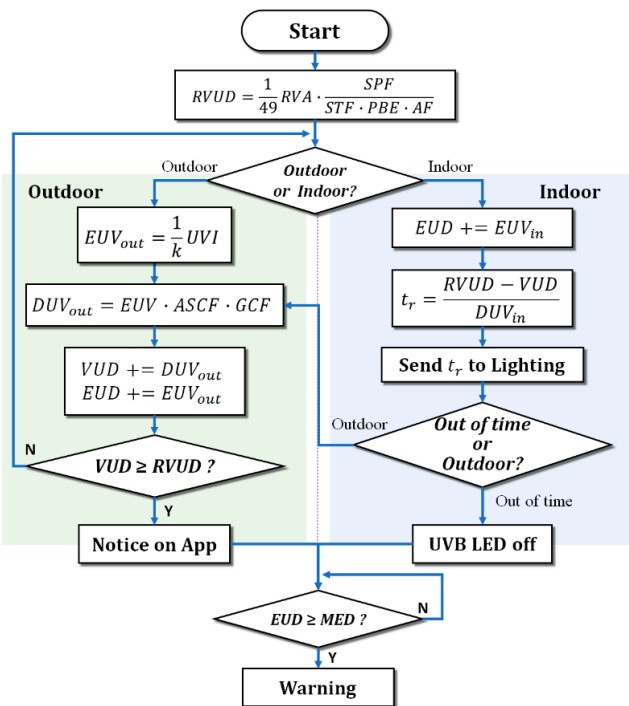

**Figure 6.** UVB LED general lighting and block diagram of UVB LED general lighting.

In the UV exposure amount calculation algorithm, firstly, the daily Recommended Vitamin D UV Dose (*RVUD*) required for the users was calculated. For that, Equation (3) was derived to calculate *RVUD* using the calculation in Equation (2) of the estimated vitamin D synthesis amount (*VitD*) proposed in the study by Godar. In Equation (2), the *STF* is the skin type factor. In the case of skin type II, the STF is 3.2/3, for skin type III, it is 3.2/4, for skin type IV, it is 3.2/5.25, and for skin type V, it is 3.2/7.5 [23]. *PBE* is the percentage of body exposure upon referring to the Lund and Browder chart and differs according to the season and user's age [23,24]. *AF* is the age factor and has a value of 1.0 for the user aged lower than 21 years old, 0.83 for the user older than 22 and younger than 40, 0.66 for the user older than 41 and younger than 59, and 0.49 for the user older than 60 [25]. Lastly, *SPF* is the sun protection factor which equals 1 if sunscreen is not used.

$$VitD \; [\text{IU}] = 49 \; VUD[\text{J}/\text{m}^2] \frac{STF \cdot PBE \cdot AF}{SPF}, \tag{2}$$

$$RVUD[\text{J}/\text{m}^2] = \frac{1}{49} RVA \; [\text{IU}] \frac{SPF}{STF \cdot PBE \cdot AF}. \tag{3}$$

In order to calculate *RVUD* by individual in Equation (3), information of the daily individual recommended vitamin D amount (*RVA*) is needed and is differentially applied according to the user's age. In this paper, the daily recommended vitamin D intake standard suggested by the Korean Ministry of Health and Welfare (KMOHW) is applied [26] and *RVA* by each user as 200 IU under 11 years old, 400 IU for under 11 years old and 65 years old, and 600 IU for over 65 years old. After calculating the *RVUD*, the cumulative UV dose was calculated by exposure to the sunlight in case the user is outdoors after judging the current location of the user. The status of satisfaction for the *RVUD* was informed to the user. Erythemal UV outdoor (*EUV*$_{out}$) was calculated from Equation (1), and Erythemal UV Dose (*EUD*) was calculated by accumulating *EUV*$_{out}$ per second. In addition, Vitamin D producing UV radiation (Vitamin D UV Dose, *VUD*) was calculated by accumulating Vitamin D UV irradiance outdoors (*DUV*$_{out}$) over time. The equation to obtain *VUD* is as shown in Equation (4).

$$\begin{aligned} VUD &= \int_0^{t_c} DUV(t) \; dt, \\ &\cong \sum_{t=0}^{t_c} DUV(t) \cdot \Delta t = \sum_{t=0}^{t_c} DUV. \end{aligned} \tag{4}$$

Equation (4) was used to obtain *VUD* from the moment of measuring *EUV* until current time $t_c$. Here, $\Delta t$ is for 1 s, the *EUV* was measured every second with the IoT UV measurement device, and the *VUD* could be calculated by Equation (4) using the numerical integration method. *DUV*$_{out}$ was measured using Equation (5).

$$DUV_{out} = EUV_{out}[\text{W}/\text{m}^2] \cdot ASCF \cdot GCF. \tag{5}$$

In Equation (5), the calculation of the *DUV*$_{out}$ includes an action spectrum conversion factor (*ASCF*), a geometric correction factor (*GCF*) that converts the UV radiation to the surface of the body into radiation intensity for the body surface by modeling the human body in a cylindrical form. This value was changed according to the latitude and season of the user while the *ASCF* by season and *GCF* reference values in Table 1 proposed by Pope were applied [27,28].

**Table 1.** Action spectrum conversion factor (*ASCF*) and geometric correction factor (*GCF*) tables.

| Latitude | Item | Seasonal Conversion Factors | | | |
| --- | --- | --- | --- | --- | --- |
| | | **Spring** | **Summer** | **Fall** | **Winter** |
| 30° N | *ASCF* | 0.742 | 1.11 | 1.061 | 0.91 |
| | *GCF* | 0.593 | 0.593 | 0.655 | 0.655 |
| 35° N | *ASCF* | 1.049 | 1.104 | 1.029 | 0.842 |
| | *GCF* | 0.600 | 0.600 | 0.655 | 0.655 |
| 40° N | *ASCF* | 1.008 | 1.067 | 0.963 | 0.7 |
| | *GCF* | 0.608 | 0.608 | 0.681 | 0.681 |

If the location of the user is the indoor environment where the UVB LED general lighting of the proposed system is installed, the irradiation time of the UVB LED lighting is calculated based on the *VUD* that has been irradiated so far and sent to the proposed lighting. The irradiation time of the UVB LED $t_r$ is based on Equation (6) below.

$$t_r = \frac{RVUD - VUD \ [\text{J/m}^2]}{DUV_{in} \ [\text{W/m}^2]} \ [\text{s}]. \tag{6}$$

$t_r$ can be obtained as a second unit by dividing the deficit of the daily optimal UV dose by the Vitamin D UV irradiance indoors ($DUV_{in}$). Here, $DUV_{in}$ is obtained as shown in Equation (7) by applying the vitamin D production weighting function (action spectrum for the production of Vitamin D, $W_d$ ($\lambda$)), which is a function of the wavelength-based weighting value that promotes the vitamin D production [29]. The measured values in Section 4.1 were used.

$$DUV_{in} = \int_{252}^{330} E(\lambda) \cdot W_d(\lambda) \ d\lambda. \tag{7}$$

## 4. Experiment and Simulation

In order to support individual daily UV dose through the proposed system, accurate measurement of UV exposure outdoors and provision of safe UVB dose through indoor UVB LED lighting is essential. Therefore, the measurement accuracy of the IoT UV measuring device constituting the proposed system was evaluated and experiments were conducted to confirm the radiation characteristics of UVB LED lighting. Furthermore, simulations were conducted to confirm that the proposed system could meet individual daily UV dose and support vitamin D synthesis.

### 4.1. Performance Test for IoT UV Smart Device and UVB LED Lighting

IoT UV measurement devices using UVI sensors must be calibrated through comparative experiments with the reference equipment. Their performances can be improved by applying the correction formula that reflects regional characteristics and seasonal characteristics [16]. For the calibration and performance evaluation of the IoT UV measuring device, UVI from the noon (12 p.m.) until the sunset on 21 February 2019 at the latitude of 36.85 and the longitude of 127.15 (Kongju National University, Korea) was used. A spectrometer (CAS 140 CT-152, Instrument Systems, Munich, Germany) was used as a reference device for the calibration and performance evaluation of the measuring instrument. In addition, a tracking system that follows the sun in the vertical direction by calculating the azimuth and altitude of the sun based on the latitude, longitude, and date of the current location was installed. The correction formula by linear regression analysis was derived based on the UVI value acquired through the IoT UV measurement device and the reference device at the same time for the natural light. The correction formula was applied to the IoT UV measurement device. Figure 7 shows the experimental results of the reference device for the sunlight and the IoT UV measuring device. The measurement results of the IoT UV device are displayed before and after calibration together. The mean absolute percentage error (MAPE) of the data measured by

the spectrometer and the IoT UV measurement device before calibration was 45.5%, but after applying the calibration formula, the MAPE could be decreased to about 11.1%.

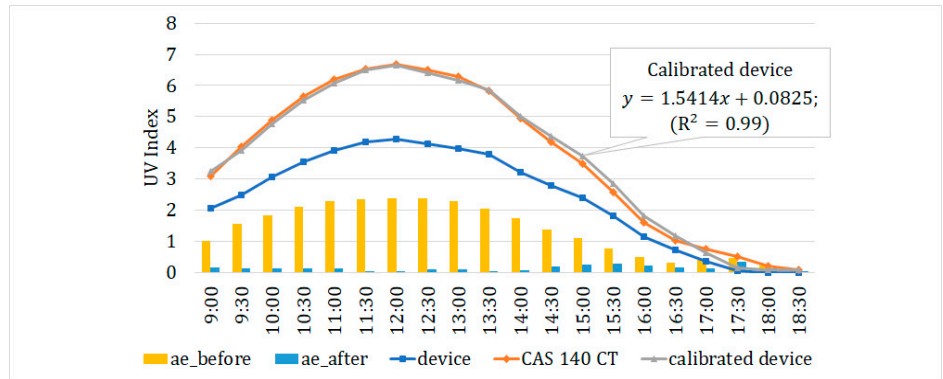

**Figure 7.** Performance test results of the measuring device.

An experiment to check the characteristics of UVB LED general lighting was also conducted. The characteristics of light were measured using the spectrometer (CAS 140 CT-152, Instrument Systems) in the lighting box having a dimension of 120 (w) × 120 (d) × 200 (h) cm to which outer light does not penetrate. The UVB LED general lighting was fixed at the top of the lighting box, the spectrometer was installed at the bottom, and the distance between the lighting and spectrometer was set to 150 cm considering the installation condition of the general lighting. In addition, the input current of 201 mA was applied in the UVB LED module considering safety in the UV zone of the photobiological safety standard of the lighting (IEC 62417) [30,31]. Figure 8 shows the measurement results of the UVB LED general lighting. Irradiance for the UVB, UVA, and visible areas of the proposed lighting was measured as 1.65, 9.20, and 2560.09 mW/m² in Figure 8a, and the illuminance of the proposed lighting was measured as 509 lux. Therefore, it is assumed that spectral irradiance was implemented for almost all the visible areas. In order to confirm the photobiological safety of the proposed lighting for the UV region, the Near UV (NUV) of 1.4885 mW/m² was obtained by integrating the wavelength of 315–400 nm. Figure 8b shows the enlarged UV region. Figure 8c shows the result of applying the Actinic weighting function to the UV region. The Actinic UV irradiance (AUV) was 0.9969 mW/m² after applying the weighting function to the wavelength band of 200–400 nm. This confirms that the proposed lighting meets the NUV 10 W/m² and AUV 1 mW/m², which is the safest exempt standard of the photobiological safety standard of the lighting (IEC 62471). As a result of applying Equations (7) and (1), it was confirmed that $DUV_{in}$ and EUV of the proposed lighting had the values of 2.5089 and 4.832 mW/m², respectively.

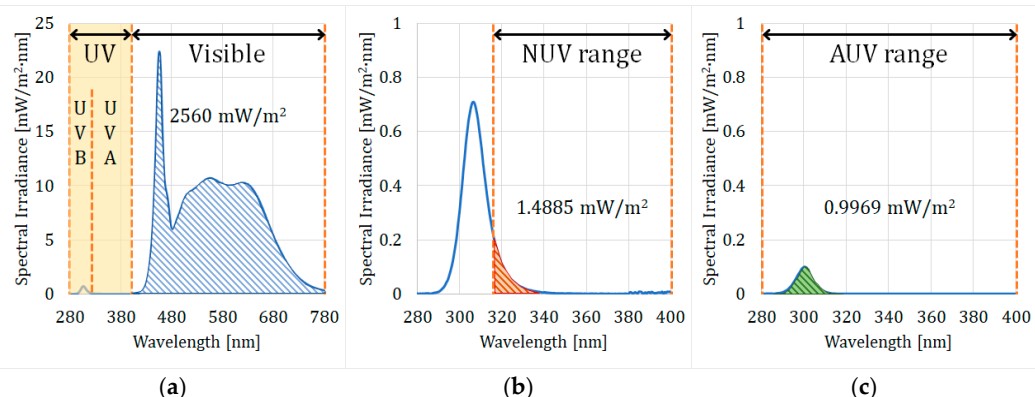

**Figure 8.** Measurement results of the UVB LED general lighting using a spectrometer (CAS 140 CT-152, Instrument Systems). (**a**) Spectral irradiance of the proposed lighting; (**b**) spectral irradiance of UV region and Near UV (NUV) irradiance; (**c**) spectral irradiance of UV region with actinic hazard weighted and Actinic UV (AUV) irradiance.

*4.2. Lighting System Simulation*

　　The service execution results of the proposed lighting system were verified by a user's outdoor activity type simulation based on the actual UV data of a natural light of 4.1. At this time, it was assumed that a user started outdoor activities at 9 a.m., where UVI was near the average UVI per day, and performed outdoor activities for 30, 20, 10, and 0 min, respectively. When the user who carried out outdoor activities entered the indoor environment, the indoor UVB LED lighting was provided and the irradiation time of the UVB LED lighting was calculated considering the UV exposure amount of the user. For the simulation, the *RVUD* for each age according to *PBE* was calculated using Equation (4) to define the individual characteristics of the user as shown in Table 2. At this time, the skin type was set to 2 or 3, which is the common skin type of Asians, and four cases were considered according to the classification step of the *AF*.

**Table 2.** Age-specific Recommended Vitamin D UV Dose (*RVUD*, skin types 2 or 3) according to the exposure area of the user.

| Experiment Cases | Age | *AF* | *RVA* (IU) | *RVUD* According to *PBE* ($J/m^2$) | | | | | |
|---|---|---|---|---|---|---|---|---|---|
| | | | | Skin Type II | | | Skin Type III | | |
| | | | | 10% | 15% | 30% | 10% | 15% | 30% |
| CASE 1 | 0–21 | 1 | 400 | 77 | 51 | 26 | 102 | 68 | 34 |
| CASE 2 | 22–40 | 0.83 | 400 | 92 | 61 | 31 | 123 | 82 | 41 |
| CASE 3 | 41–59 | 0.66 | 400 | 116 | 77 | 39 | 155 | 103 | 52 |
| CASE 4 | 60– | 0.49 | 600 | 234 | 156 | 78 | 312 | 208 | 104 |

　　Considering age, skin type, and exposed area, the *RVUD* required for the day is about 26–312 $J/m^2$ in the case of Asians as shown in Table 2. The outdoor UV dose (*VUD*$_{out}$), the recommended exposure time to the lighting ($t_r$), the satisfactory levels of UV dose to be provided indoors (*VUD*$_{in}$), and *RVUD* are shown in Table 3.

**Table 3.** Comparison of UV exposure amount according to the duration of outdoor activities of the user.

| Outdoor Time (EUD) ($J/m^2$) | Factors | Skin Type II (Unit: $J/m^2$) | | | | Skin Type III (Unit: $J/m^2$) | | | |
|---|---|---|---|---|---|---|---|---|---|
| | | CASE 1 | CASE 2 | CASE 3 | CASE 4 | CASE 1 | CASE 2 | CASE 3 | CASE 4 |
| | *RVUD* | 51 | 61 | 77 | 156 | 68 | 82 | 103 | 208 |
| 30 Min (157.38) | *VUD*$_{out}$ | 95.90 | | | | 152.38 | | | |
| | $t_r$ (s) | 0 | 0 | 0 | 12,474 | 0 | 0 | 1482 | 23,248 |
| | *VUD*$_{in}$ | 0.0 | 0.0 | 0.0 | 60.28 | 0.0 | 0.0 | 7.16 | 112.34 |
| | Total EUD | 157.38 | 157.38 | 157.38 | 188.68 | 157.38 | 157.38 | 161.10 | 215.70 |
| | Met *RVA* | O | O | O | O + I | O | O | O + I | O + I |
| 20 Min (91.40) | *VUD*$_{out}$ | 57.52 | | | | 91.40 | | | |
| | $t_r$ (s) | 0 | 816 | 4093 | 20,417 | 2174 | 5057 | 9426 | 31,192 |
| | *VUD*$_{in}$ | 0.0 | 3.95 | 19.78 | 98.66 | 10.50 | 24.44 | 45.55 | 150.72 |
| | Total EUD | 91.40 | 93.45 | 101.67 | 142.62 | 96.85 | 104.09 | 115.05 | 169.66 |
| | Met *RVA* | O | O + I | O + I | O + I | O + I | O + I | O + I | O + I |
| 10 Min (44.58) | *VUD*$_{out}$ | 28.06 | | | | 44.58 | | | |
| | $t_r$ (s) | 4752 | 6915 | 10,191 | 26,516 | 8272 | 11,155 | 15,524 | 37,290 |
| | *VUD*$_{in}$ | 22.96 | 33.41 | 49.25 | 128.13 | 39.97 | 53.90 | 75.01 | 180.19 |
| | Total EUD | 56.50 | 61.93 | 70.15 | 111.11 | 65.33 | 72.57 | 83.53 | 138.14 |
| | Met *RVA* | O + I | O + I | O + I | O + I | O + I | O + I | O + I | O + I |
| 0 Min (0) | *VUD*$_{out}$ | 0 | | | | 0 | | | |
| | $t_r$ (s) | 10,559 | 12,721 | 15,998 | 32,322 | 14,078 | 16,962 | 21,331 | 43,096 |
| | *VUD*$_{in}$ | 51.02 | 61.47 | 77.30 | 156.18 | 68.03 | 81.96 | 103.07 | 208.25 |
| | Total EUD | 26.49 | 31.92 | 40.14 | 81.09 | 35.32 | 42.56 | 53.52 | 108.12 |
| | Met *RVA* | I | I | I | I | I | I | I | I |

In Table 3, the Met *RVA* entry was marked with 'O' or 'I' or 'O + I' when *RVA* could be met in outdoor, indoor, or both. When outdoor activities are carried out for 30 min, all outdoor activities are found to meet the *RVUD* of those aged 40, although outdoor activities are not enough for users aged 60 and over, whereas the dose requirement of adults aged 41 to 59 depends on the skin type. When outdoor activities are performed for 20 min, skin type 2 is satisfied with outdoor activity until age 21, and for other ages, indoor UVB LED lighting appears to be supplemented. To achieve the daily amount of vitamin D synthesis, after 30 min of outdoor activity, users 60 years or older with skin type II or III can use the indoor UVB LED lighting for 3 h, 27 min, and 54 s (12,474 s) or 6 h, 27 min, and 28 s (23,248 s.) In addition, users who are not carrying out outdoor activities and who have skin type II can meet their daily vitamin D synthesis amount by exposure to UVB LED general lighting for at least 2 h 55 min, and 59 s (10,559 s) and up to 8 h, 58 min, and 42 s (32,322 s) of the total amount. Meanwhile, users with skin type III are able to meet the daily amount of vitamin D synthesis when exposed for at least 3 h, 54 min, and 38 s (14,078 s) and up to 11 h, 58 min, and 16 s (43,096 s). The total *EUD* in the indoor and outdoor areas were found to be a minimum of 26.49 J/m$^2$ and a maximum of 215.70 J/m$^2$, and it was confirmed that these values did not exceed MED in all the cases. The simulation results in Table 3 confirm that it is possible to provide optimum UV exposure through indoor UVB LED general lighting for all ages and skin type users.

## 5. Conclusions

In this paper, a UVB LED general lighting system based on a UV exposure calculation algorithm that allows the user to meet the required UV dose per day is proposed. The proposed system was composed of an IoT UV measurement device that measures the UVI of outdoor natural light, a smartphone that provides information related to UV and vitamin D, a control function of UVB LED lighting, and UVB LED general lighting that provides a safe UVB dose to the user indoors. First, the IoT UV measuring device calculated the intensity of the $EUV_{out}$ by measuring the UVI outdoors in real time, and then sending the result to the smartphone. The smartphone then calculated the cumulative UV dose of the user who was irradiated outdoors based on the EUV value received from the IoT UV measuring device, comparing the cumulative *DUV* dose and the *RVUD* for the user for one day. If the cumulative UV dose outdoors did not reach the *RVUD*, then the indoor UVB LED lighting provide the deficient UV dose. In this case, the *RVUD* by individual was calculated by reflecting individual characteristics such as age, skin type, and exposed area, and then, based on the outdoor UV exposure amount, the control time of the indoor UVB LED lighting was calculated. After that, the control information of the lighting was constituted as packets and sent to the UVB LED lighting, thereby realizing the function of providing the UV dose through the indoor UVB LED lighting. At this time, UVB LED general lighting was developed by applying four UVB LED light sources of 20 mW radiant flux and a current of 201 mA. In order to evaluate the performance of the proposed lighting system, a comparative experiment using a spectrometer was performed. Experimental results of natural light UVI showed the MAPE could be decreased to about 11.1% compared with the spectrometer. In addition, in the experiments conducted in the lighting box to which external lighting was blocked, UVB LED lighting was estimated to be 0.9969 and 1.4885 mW/m$^2$ for AUV and NUV, respectively, which satisfied the safest level (exempt) when it was evaluated with the risk to the ultraviolet region of the photobiological safety standard. In the simulation for confirming the operation performance of the proposed lighting system, the users of all age groups with skin types 2 and 3 could be provided the daily required UV dose when the outdoor activities are performed for 0–30 min on a day when the average UVI is 3.

In the future, additional experiments for service in consideration of multiple users in the same space and for operation in different regions and climatic conditions are needed for the commercialization of the proposed lighting. In addition, for users who want to meet the appropriate UV exposure for a short period of time, additional research is needed to develop UVB LED special lighting for therapeutic purposes other than the general lighting type and to develop a lighting system linking to UVB LED

special lighting. Furthermore, animal experiments are planned to check the supporting function of vitamin D synthesis through the proposed lighting.

**Author Contributions:** Methodology and writing of the original draft: D.-H.P. Investigation and writing of the original draft: S.-T.O. Conceptualization and supervision: J.-H.L.

**Funding:** This work was supported by the National Research Foundation of Korea (NRF) grant funded by the Korea government (MSIP) (No. 2017R1A2B2005601). The research was supported by the International Science and Business Belt Program through the Ministry of Science and ICT (2015-DD-RD-0068-04).

**Conflicts of Interest:** The authors declare no conflict of interest.

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
