# Peer review of "Development of UVB LED Lighting System Based on UV Dose Calculation Algorithm to Meet Individual Daily UV Dose"

_applsci, doi:10.3390/app9122479_

Round 1

Reviewer 1 Report

Dear Authors,

I have read with great interest your paper. Basically you propose some mean to provide a mean in order for people to monitor and enhance their production of D vitamin. The context and device description are adequately developed. The results seems promising.

I would like you to rework some points:

1) never express a value with  5 or 6 digits. There are physic constants and a whole calibration and uncertainty chain. For instance, rounding table 15 values to integer would be adequate. Does 77 makes such a big difference from 76.5306 ? The readability will be greatly enhanced

2) The introduction start by exposing a few reasons why daylight exposure is beneficial. I would like you to go one step further. The first sentence should read: "Proper exposure to daylight is essential for physical and mental health because ... [1] The idea is to indicate one or two reasons sustaining  the argument.  Furthermore, I would like you to properly balance advantages and drawbacks of UV. Among the drawbacks: DNA strong absorption at 260 nm; two of the DNA bases also absorb at 280 nm. So IF you use UVB, you should have nothing between 280 and 290 nm. Graph 8 should include a zoom with logarithmic scale  of the region 280 - 400 nm

3) As the final target is to use your system on humans, I have an ethical concern about your research: you should include one medical doctor in the author team. It's not about simulating some lighting system, it's about people. Your system has to be validated in a medical framework; and this validation has to be integrated into the research project, right from the beginning.

4) The whole paper is sometimes a bit verbose, with f.i. complex words appearing in two consecutive sentences. Try to use pronouns to shorten the sentences

5) I'm not sure giving the packet structure helps a lot. Lines 137 to 157 should be reduced. You implemented some robust protocol over BLE and that's it.

6) Sect 4.1. You give the location coordinates but not their referential; nor the observation direction. Are both devices pointing upwards or tracking  the sun ? What about cloudiness ? In eq. 9, you should also provide some confidence interval for both coefficients. Fig 7B legends are difficult to read

7) From Table 6, it seems elderly people needs to spend between 4 and 14 hours under artificial UVB sources ? Is it a reasonable result ? It is doable ? Is it correlated with studies of D vitamin deficiency in elderly peoples ?

8) what about the energy consumption of the UV LEDs ?

9) Ref 12 (Klepeis) is a bit outdated. There should be more recent publications about the increasing tendency to stay inside. There was no smartphone in 2001.

Author Response

I sincerely thank the reviewer who reviewed my paper.

I have attached an answer to the reviewer.

Reviewer 2 Report

- bad qualities of pictures, some are even not readable

- there are minor language/ grammatical problems

- the titles of the pictures and tables are not correctly arranged

- many abbreviations are without full names (e.g. UVI, loT, XOR when indescriptions, and so on), they are confusing

- table 1 does not contain 16 bytes' data as described in the article

- Figure 2, what is the different between solid arrows and dashed arrows?

- this paper tells the effect of lacking exposure to UVB, how about the effect when over exposure to UVB? For instance, when a user does some outdoor activities in the morning and come back indoors at noon, how does the device and algorithm judge if the UVB LED should be turned on or not? Especially when it is unsure if the user may do some more outdoor activities in the afternoon. In this case, when the device turns the LED on, it may cause over exposure. (This can be implemented within the last chapter as an outlook.)

Author Response

(The authors gave the same response as above.)

Reviewer 3 Report

Dear authors, while treating an interesting topic, to me it is not clear which are the specific goals of the paper and which are the novelty or relevant points developed.

I could say that a specific application architecture of a wearable device that is call to interact with one/some luminaires. Further on is presented the calculation procedure implemented developed by Godar, D. E.; Pope, S. J.; Grant, W. B.; Holick, M. F.  Then a calibration and an evaluation procedure of a specific implementation (that is not properly describes) is done and finally an analysis of time of use for this product based on very short outdoor exposure time is presented.

As given, an specific product development is presented (that is not compared to any other possible alternative in hardware or calculation procedure or that has an proper detail design analysis to obtain an optimized solution) and it performance is analyzed again without any further analysis of comparison or critical review for improvement. Thus, the relevance or significance to potential readers is reduced. An alternative/comparative study from a third party is impossible, as the construction detail of the solutions (power, LED packages, drivers, control systems…) are not given.

However, some technical information is sometimes given without a specific reason of further details. For example: “The UVB LED (LG InnoTek, Korea) with a specification of 20mW output was implemented at the four corners to provide UVB” & “the input current of 201 mA was applied in the UVB LED module considering safety in the UV zone of the photobiological safety standard of the lighting (IEC 62417) [28-29]”

Which specific LED has been used? If it is a 20mW and the input current is 201 mA. The Vf is 0,1V? How far from the nominal is the polarization point? Is there an array of series/parallel LEDs? Of how many? How far are you from overpassing the extent grade?

Why such a big range of illuminance performance is given: “was developed so that an illuminance of 300-600lux can be realized at a distance of 3750px from the ceiling”. This depends on what?

Why “warm white and cool white LED chips are located”? Figure 8 seems to indicate that only warm LEDs are turned on along with UV emitters.

The description of the architecture of the system presented is quite simple and without any significant innovations as there are already many measurement wireless/wearable sensor developed. Too many effort is given to detail elements that are non-relevant or introduce little advance or novelty such as the communication protocol. Moreover, the system presented could be describes as inefficient as many non-information bytes are used and a permanent package emission is done wasting a lot of valuable energy for a battery dependent system.

Several tests to find out technological limits or possibilities could be of much more interest. For example, to estimate in which way UVB emissions can be increased still being in the exempt grade according to the IEC 62417 in order to minimize to indoor exposure or at least decrease the long lasting required time presented

Finally, in general, precision and clearness must be improved:

One significant incongruence: Values presented in chapter 4.1 (Line 258) and in the conclusions (Line 350) do not seem to match

“The measurement results before and after applying the correction formula in Equation (9) is as shown in Figure 7. The mean absolute percentage error (MAPE) of the data measured by the spectrometer and the IoT UV measurement device before calibration was 45.5%, but after applying the calibration formula of Equation (9), the MAPE could be decreased to about 11.1%.”

“Experimental results of natural light UVI showed that the IoT UV measurement device showed 93.4% measurement accuracy compared with the spectrophotometer”

Some other technical errors:

·       In the text it is changed between the mention of UV, UVA and UVB without a clear pattern Moreover, as described in the introduction, it seems that only UVA are harmful for the human body and that UVB presents only benefits.

·       Table 5 present no units on their parameters indicated

·       Page 1 Line 42. Some acronym are firs used (such as RVA) and further on presented (page 6 Line 200)

·       Some references are not indicated (Line 190): “For that, Equation (4) was derived to calculate a daily optimum UV dose (RVUD) using the calculation in Equation (3) of the estimated vitamin D synthesis amount (VitD) proposed in the study by Godar.”

·       Line 127. “here, k=40” Why?

·       Line 136 and Line 207. The concept of “real time” in data processing or communication specifications is not properly used

·       National standards as the KS A 3011 should not be used if another international documents have been developed in order to improve soundness within a wider range of readers

·       Figure 7 includes text that is unreadable

English writing presents many significant mistakes and error. A deep review is required.

Some phases lack of sense. Several examples:

(Line 61): “However, these UV lighting technologies are types that control the irradiation time of lighting among the users, but they fail to realize a lighting system providing short UV dose indoors by linking the UV exposure amount individually outdoors.”

(Line 90) “In the mobile device, the information of EUV dose is obtained through outdoor activities and the consequent vitamin D synthesis amount to the users”

(Line 116) “The implemented UVI sensor was from Tocon E2 model from S company that gives output voltage according to the strength of the UV index. In addition, Arduino Nano (ATmega328) was implemented for the MCU. The packet for communication was then constructed whereas sending function through BLE module (HM-10) was realized.”

The same words are repeated constantly very close to each other:

(Line 92): “In addition, the presence of UVB LED indoor general lighting is checked through continuous scanning. If UVB LED indoor general lighting is detected, it is made to provide UVB dose in shortage after recognizing the user that enters the room.”

(Line 228): “which is a function of the wavelength-based weighting value that promotes the production of vitamin D production”  

Wrong verb forms and lack of articles:

(Line 131) “The byte 1 at the starting and ending were set as STX (0x02) and ETX (0x03), respectively, and each data contained in the packet were (WAS) categorized as (A) NULL (0xFF) byte.”  à “The (starting and ending BYTES) were set as STX (0x02) and ETX (0x03), respectively, and each data contained in the packet (WAS) categorized as (A) NULL (0xFF) byte.”

For all the reasons presented, I consider that the paper requires a mayor revision before being suitable for publishing.

Sincerely

Author Response

(The authors gave the same response as above.)

Round 2

Reviewer 1 Report

Dear authors, I was amazed to read the other reviewers comments. The spotted issues were quite similar. They can be categorized as: 1) minor points like graphs readability, sentences needing reworking, ... Those points were addressed in the revised version 2) major points: a) the communication protocol between smartphones and control system is not new nor innovating. All the elements were presents in the '80s. Bandwidth and energy are lost because the protocol is too simple. As this protocol is just some ancillary part, its description should be skipped. b) the validity of the proposed scenarii are not assessed. While I ask 'is it OK for some person to stay 14 hours under your system', the answer is vague and does not address the main concern: can you ask one person to stay in the same room for 14 hours a day ? What are the pros and cons of such situation ? c) There is still a major ethical concern. Telling that the protocol will follow IEC 62471 standard does not bring any reassurance to me, if there is no proper medical validation and follow-up. So I reinstate my requirements: include someone with a biomedical degree in the author team or ensure supervision by a dermatology unit. To conclude, I MAINTAIN my vote of "Need major rework" as I am still not convinced that the major issues were properly addressed, and I give you ONE more chance to correct them. Regards Pascal Dupuis

Author Response

Dear Reviewer.

I sincerely thank the reviewer who reviewed my paper.

The answers and corrections to your points are arranged the attached file.

Sincerely,
SeungTaek Oh.

Reviewer 3 Report

Dear Authors,

Some elements indicated on the previous reviews have been corrected such as the introduction, that has improved from the first versions.

However other ones still need a deep review in order to be published:

Result given are somehow inconclusive and need a deeper analysis. For example:

Line 279 “As a result, AUV and NUV, which are the UV risk items of the photo biological safety standards (IEC 62471) of the lighting, are 0.9969mW/m2 and 1.4885mW/m2, respectively, which fall under the safest grade (exempt).” It could be important to indicate where is the limit of the exempt grade, to know how close it is from being somehow harmful. These values as presented give no significant information.

“Meanwhile, the calculated EUV and DUVin are 2.5089 and 4.832mW/m2.” Is this good or bad?  High or low? Which relevance does it has?

The point of the longtime of exposure requirements in neither treated. Does it implies any other consequence to the health to the people? “Afterwards, animal experiments should be carried out to confirm the performance of the proposed lighting system, whereas a validation of the effect of the supporting vitamin D synthesis through UVB LED illumination should also be conducted” Should not this be done in the first place. This is something gives as true on the paper and now the consequence expected do not seem to be granted.

Why is it necessary to “development of special purpose UVB LED for therapy other than general lighting type.”? Are the UV LED used not adequate? In which case are UVB LED used in general lighting?

The explanation of the communications protocol still presents or offers no relevant information. Your own conclusions do not even mention that part of the paper at all. It is not clear the main area of science under investigation as they are mixed (but not related) many different disciplines (treated as independent islands)

The relevance of a calibration process of a specific sensor neither seem to be too significant at the level presented.

The most relevant part of the paper (chapter 4 and 5) present very little changes to solve the considerations presented, and the further details given presents more doubts than certainties.

Also, there are still a large number of writing and formal mistakes:

There is a subject something missing in the expresion: ”UVB LED indoor general lighting” at the end it could include the word “luminaire”. Also in others, like “based on seasonal” trends o patterns maybe.

It still presents several time the same word too close in the text: “the result to the smartphone. In the smartphone, cumulative UV” and Line 104 “UVI information to the smartphone. The smartphone was realized for calculating”

In some cases 1 sentence is repeated completely: Line 135 “The protocol for real-time transmission is shown in Table 1 and consists of 16 12 bytes in total. The protocol for real-time transmission is shown in Table 1 and it consists of 12 bytes in total.”

There are two figures 5 and no figure 4.

Some Legend on figure 7(b) are still impossible to be read. A segment of the image seems to be repeated.

Figure 8 must be reviewed. It repeats the same information several times with graph modules that are also equal. The differences between the cool and warm light should be explained. In order not to confuse readers, scale units of the vertical axis of the different figures should be the same. Control of changes seems to be generating duplicities.

Please make an additional effort to improve the concerns presented

Sincerely

Author Response

(The authors gave the same response as above.)

Round 3

Reviewer 1 Report

Dear authors,

I congratulate you for the efforts you put in reviewing your paper, removing points of contention. Now I feel confident about your ideas being further tested on, as they are safeguards against UV danger.

I still have a few small points of attention requiring actions:

1) the last paragraph mentions animal trial; it should also contain a reference to clinical trials, as indicated in the answer to the reviewer remarks

2) pay some care to physical quantities reporting. F.i, on line 268:

a) the power is 0.995 mW/m^2 (there should be a unsecable space between the numbers and the unit, meaning the numbers may not end one line and the units start the next one)

b) the range is 315 - 400 nm. The intermediate hypen is called "em-dash" and is a bit longer than the "minus sign", and there should be one non breakable space between numbers and units.

One line is mentioned, but the lack of spacing occurs at many other places. Those em-dash and not-breakable spaces are to be find in the "typography symbols" of your text editor.

3) please mention the spectrometer brand and model which was used to produce the results of fig. 8

Regards

Author Response

I have attached my answer as a file.

Reviewer 3 Report

I first would like to recognize the effort done to improve the paper. As presented now, most of the requirements have been attended with appropriate inclusion of information or accomplishing changes of structures and presentation both in the text and in the figures.

As presented now, specific data extracted from references are used to justify the overcome procedures and the analysis developed.

Finally, concerning the ethical aspects of possible danger exposition to human beings along the test generated, security concerns and procedures used to prevent them are now presented in a clearer way. Even tough they can be criticized depending on the level of requirements that may be applied from different points of view, in my opinion it is can be considered that it is now analyzed well enough as to be published.

As the only requirement that I would like to make at this point of the review process along with the recommendation for it to be publishing, and recognizing again an improvement from the first text presented, still a slightly deeper format and language revision must be still conducted. Some examples of sentenced with correction needs:

·       References has some cases format mistakes. One case: “5. Shim, J. H. Anti-aging Effects of P7C3 in UVA-irradiated Human Dermal Fibroblasts. Asian 366 J Beauty Cosmetol. 2017, 15.1, 45-53.”

·       Line 133 “UVB LED indoor lighting was developed to a general lighting type so that users can receive UVB 134 dose naturally while indoors. Figure 4 shows UVB LED indoor lighting and its block diagram.”

·       Line 139 “The Warm/Cool LEDs … were alternately arranged in 16 horizontal and 18 vertical lines to control the brightness and render color temperature control possible”

·       Line 250 “Figure 7 is the experimental result for the sunlight and the results before and after the correction of IoT UV measuring device are also shown together.”

Sincerely

Author Response

I have attached my answer as a file.
